# Cytoreductive Surgery (CS) with Hyperthermic Intraperitoneal Chemotherapy (HIPEC): Postoperative Evolution, Adverse Outcomes and Perioperative Risk Factors

**DOI:** 10.3390/healthcare13070808

**Published:** 2025-04-03

**Authors:** Lucía Valencia-Sola, Ángel Becerra-Bolaños, María Mateo-Ferragut, Virginia Muiño-Palomar, Nazario Ojeda-Betancor, Aurelio Rodríguez-Pérez

**Affiliations:** 1Department of Anesthesiology, Intensive Care and Pain Medicine, Hospital Universitario de Gran Canaria Doctor Negrín, 35010 Las Palmas de Gran Canaria, Spain; lvalsol@gobiernodecanarias.org (L.V.-S.); vmuipal@gobiernodecanarias.org (V.M.-P.); nojebet@gobiernodecanarias.org (N.O.-B.); arodperp@gobiernodecanarias.org (A.R.-P.); 2Department of Medical and Surgical Sciences, Universidad de Las Palmas de Gran Canaria, 35001 Las Palmas de Gran Canaria, Spain; maria.mateo107@alu.ulpgc.es

**Keywords:** cytoreductive surgery, HIPEC, intraoperative chemotherapy, peritoneal carcinomatosis, perioperative management, postoperative complications, mortality

## Abstract

**Background:** Cytoreductive surgery (CS) and hyperthermic intraperitoneal chemotherapy (HIPEC) increases survival in peritoneal carcinomatosis, but complications may affect the long-term prognosis. We aimed to evaluate the postoperative evolution after CS + HIPEC, the appearance of adverse outcomes, and the associated risk factors. **Methods:** This was a retrospective observational study evaluating clinical practice in patients undergoing CS + HIPEC from 2016 to 2023 in a tertiary-level university hospital. The pre-, intra-, and postoperative variables were collected. The postoperative evolution, the appearance of postoperative complications, and the mortality were analyzed according to the perioperative data. **Results:** In total, 62.3% of the patients developed some kind of complication. Renal failure was related to the length of surgery [mean difference (md) 111 min, 95% CI 11–210, *p* = 0.029], postoperative vasoactive support [Odds Ratio (OR) 3.4, 95% CI 1.1–10.6, *p* = 0.033], and non-invasive mechanical ventilation (OR 5.5, 95% CI 1.5–20.5, *p* = 0.007). Respiratory failure was associated with renal replacement therapies (OR 13.8, 95% CI 1.3–143.9, *p* = 0.006), postoperative creatinine (md 0.27 mg·dL^−1^, 95% CI 0.1–0.4, *p* = 0.001), and C-reactive protein (md 33.5 mcg·L^−1^, 95% CI 0.1–66.8, *p* = 0.049). Infectious complications were related to the length of surgery (md 84 min, 95% CI 12–156, *p* = 0.024), non-invasive mechanical ventilation (OR 4.4, 95% CI 1.2–16.1, *p* = 0.018), and renal replacement therapies (OR 11.6, 95% CI 1.1–119.6, *p* = 0.012). The hospital stay was longer in patients with complications (md 14.8 ± 5.5 days, 95% CI 3.8–25.8, *p* = 0.009). The mortality rate at 12 months was 15.6%. The mortality risk factors were the preoperative hemoglobin (md −1.7 g·dL^−1^, 95% CI −2.8–−0.7, *p* = 0.001) and creatinine (md −0.12 mg·dL^−1^, 95% CI −0.21–−0.04, *p* = 0.007) and the postoperative hemoglobin (md −1.15 g·dL^−1^, 95% CI 0.01–2.30, *p* = 0.049) and C-reactive protein (md 54.6 mcg·L^−1^, 95% CI 18.5–90.8, *p* = 0.004). Intraoperative epidural analgesia was found to be a protective factor for 12-month mortality (OR 0.25, 95% CI 0.07–0.90 *p* = 0.027). A multivariate analysis performed after a univariate analysis showed that the only risk factor for overall mortality was not using intraoperative epidural analgesia. **Conclusions:** CS + HIPEC led to a high incidence of postoperative complications, but the occurrence of complications did not seem to affect postoperative survival.

## 1. Introduction

Peritoneal carcinomatosis (PC) is the end stage of tumor dissemination from colorectal, gastric, ovarian, mesothelioma, or peritoneal tumors. Peritoneal dissemination of the tumor is associated with a poor prognosis and a lower survival rate than in those without peritoneal metastases [1]. However, the final prognosis depends mainly on the origin of the primary tumor [2]. Cytoreductive surgery (CS) and hyperthermic intraperitoneal chemotherapy (HIPEC) has demonstrated an increase in the long-term overall patient survival [3]. Although this treatment is often associated with short-term postoperative complications such as anastomotic leaks, bleeding, peritonitis, ileus, wound infection, pancreatitis, intestinal fistula, urinary tract infection, sepsis, or hematological toxicity [4], the postoperative morbidity is deemed acceptable given the improvement in oncological prognosis [5]. However, the appearance of postoperative complications may affect the initiation of other adjuvant therapies necessary to improve the long-term prognosis [6]. In addition, the aggressiveness of this treatment makes it unsuitable for certain types of patients, due to longer hospital stays and worsening conditions.

The baseline characteristics of patients, as well as their functional status, the skill of the surgical team, the distribution of the disease within the abdominal cavity, and the characteristics of the ICU are some of the factors that influence the overall success of the technique [7]. Preoperative frailty increases postoperative complications and mortality [8], especially in these highly aggressive surgeries [9]. Therefore, the role of these therapies still requires an evaluation of the consequences [7]. Preoperative optimization of the patient undergoing CS + HIPEC reduces the hospital stay, the manifestation of postoperative complications, and the readmission rate [10]. Among other recommendations, routine preoperative counseling is encouraged, with the cessation of smoking and alcohol habits one month before surgery, physical exercise, nutritional and protein supplementation in cases of malnutrition, correction of anemia, evaluation of cardiovascular risk, detection of frailty, and preoperative administration of prophylaxis for postoperative nausea and vomiting [11]. However, the application of ERAS protocols is still low in this type of surgery [12]. Therefore, patients with a serious illness undergoing a highly aggressive surgery often arrive at the operating room without having been fully optimized.

This study evaluated the postoperative evolution outcomes and the appearance of adverse outcomes after CS + HIPEC and identified the associated risk factors.

## 2. Materials and Methods

This retrospective observational study evaluated the routine clinical practice in all patients undergoing scheduled CS + HIPEC for more than 7 years in a tertiary-level hospital. After the approval of the Ethics Committee (CEI/CEIm HUGCDN #2019-357-1), we included patients scheduled for CS + HIPEC from March 2016 to December 2023. Patients with a peritoneal cancer index (PCI) higher than 20 were excluded. This manuscript adheres to the Strengthening the Reporting of Observational Studies in Epidemiology (STROBE) Statement [13] and the Declaration of Helsinki (WMA).

The following data were recorded: age; gender; weight; body mass index; ASA physical status; comorbidities (arterial hypertension, diabetes mellitus, chronic kidney disease, heart and lung diseases); preoperative hemoglobin and creatinine; diagnosis leading to surgery; length of surgery; and intraoperative fluid therapy, vasoactive support, and transfusions. The postoperative hemoglobin and creatinine, C-reactive protein and lactate, as well as the length of stay in the ICU, need for postoperative vasoactive support, invasive mechanical ventilation, and renal replacement therapies were also recorded. The postoperative pain management was also documented, taking into account the analgesic protocol; the subjective pain assessment (categorized into excellent, fair, and bad); the pain assessment according to the numerical rating scale, from 1 (no pain) to 10 (maximum pain); and the postoperative complications related to pain (poor pain control, paresthesia, arterial hypotension, and pruritus). The appearance of postoperative complications was analyzed: renal failure, respiratory complications, adynamic ileus, infectious complications, hematuria, postoperative bleeding, anastomotic dehiscence, fistula, surgical wound infection, and anemia. The postoperative hospital evolution, complications, outcomes, and mortality were analyzed according to different perioperative data.

### Statistical Analysis

Data were analyzed using the SPSS program, version 24.0. Absolute and relative frequencies were used to describe the qualitative variables. To compare qualitative variables between groups, the chi-square test was used. The quantitative variables were analyzed calculating the mean and standard deviation. The normality of the data was checked with the Shapiro–Wilk test. Quantitative variables were compared between two groups using the t-Student test in the case of normal variables and the Mann–Whitney U test when the distribution of variables did not adjust to normality. When a comparison was performed among more than two groups, the ANOVA test was used in cases of normal variables and the Kruskal–Wallis test in cases in which the distribution did not adjust to normality. For assessing risk factors related to the main postoperative complications and mortalities, the Odds Ratio (95% CI) was calculated for the qualitative variables and the mean difference (95% CI) for the quantitative variables. The survival estimation was made using the Kaplan–Meier model. A *p* < 0.05 was considered statistically significant.

## 3. Results

### 3.1. Preoperative and Intraoperative Characteristics

During the study period, 80 patients were submitted to CS. From these, three patients were excluded, as their PCI was higher than 20. So, finally, 77 patients were included for the analysis. The patient characteristics and primary tumor distributions are shown in Table 1 and Figure 1.

### 3.2. Postoperative Evolution

The mean postoperative stay in the intensive care unit (ICU) was 2.3 ± 1.7 days. Immediate postoperative management in the ICU and pain management are shown in Table 2. Five patients (6.5%) required one readmission to the ICU, and one patient (1.3%) required two ICU readmissions. The mean postoperative hospital stay was 18.3 ± 29.6 days. Figure 2 shows the comparison of analgesic management according to the NRS between the groups of patients with morphine-based and epidural analgesia. No statistically significant differences were found between groups (*p* = 0.083).

#### 3.2.1. Postoperative Complications

During the postoperative period, 62.3% of patients developed some kind of complication: 31.2% had one complication; 14.3% had two different complications, while only 16.9% had more than two complications. The hospital stay of patients without complications was 9.1 ± 10.5 days, while it was 23.9 ± 35.6 days for those with complications. The hospital stay was longer for patients with at least one complication, with a mean difference (md) between groups of 14.8 ± 5.5 days (95% CI 3.8–25.8, *p* = 0.009). The most frequent complication was infection, diagnosed in 18 patients (23.4% of the total sample, 60% of all those who suffered any complication): 7 of these patients (9.1% of the total sample) had catheter-associated infections (mean time to diagnosis: 9 ± 4 days), while the rest (14.3% of the total sample) experienced abdominal cavity infections: 6 patients (7.8%) suffered intra-abdominal collections (mean time to diagnosis: 9 ± 3 days), 2 patients (2.6%) presented small-bowel fistulae (mean time to diagnosis: 7 ± 1 days), 2 patients (2.6%) showed a pancreatic fistula (mean time to diagnosis: 8 ± 1 days), and only one (1.3%) had anastomotic dehiscence (time to diagnosis: 10 days). All the patients had received amoxicillin–clavulanic acid as prophylaxis perioperatively, followed by targeted antibiotic treatment after the diagnosis of the infection. Three patients (3.9%) had abdominal wall bleeding. The aggregated postoperative complications are shown in Figure 3. The relationship between the pre- and intraoperative variables and the postoperative complications is shown in Table 3. The relationship between the postoperative variables and the main postoperative complications is shown in Table 4.

Patients who suffered from renal failure showed a longer duration of surgery (md 111 min, 95% CI 11–210, *p* = 0.029) and more use of postoperative vasoactive support (OR 3.4, 95% CI 1.1–10.6, *p* = 0.033) and non-invasive mechanical ventilation (OR 5.5, 95% CI 1.5–20.5, *p* = 0.007). Respiratory failure was associated with postoperative renal replacement therapies (OR 13.8, 95% CI 1.3–143.9, *p* = 0.006) and with higher postoperative values of creatinine (md 0.27 mg·dL^−1^, 95% CI 0.1–0.4, *p* = 0.001) and C-reactive protein (md 33.5 mcg·L^−1^, 95% CI 0.1–66.8, *p* = 0.049). Patients with infectious complications also showed a longer duration of surgery (md 84 min, 95% CI 12–156, *p* = 0.024) and a higher postoperative use of non-invasive mechanical ventilation (OR 4.4, 95% CI 1.2–16.1, *p* = 0.018) and renal replacement therapies (OR 11.6, 95% CI 1.1–119.6, *p* = 0.012).

#### 3.2.2. Postoperative Mortality

The mortality rate at 6 months was 10.4%; at 12 months, it was 15.6%; and the overall mortality at the time of this study was 33.8%. The median survival time was 63.11 months (95% CI 54.17–52.04) (Figure 4).

No relationship was detected between mortality at 6 and 12 months and renal failure (*p* = 0.542 and *p* = 0.247, respectively), respiratory failure (*p* = 0.542 and *p* = 0.702, respectively), adynamic ileus (*p* = 0.542 and *p* = 0.247, respectively), or infectious complications (*p* = 0.443 and *p* = 0.885, respectively). The relationships among the pre-, intra-, and postoperative variables and the overall mortality are shown in Table 5.

Quantitative variables significantly related to the overall mortality were dichotomized through the relevant clinical cut-off points as follows: BMI ≥ 30 kg·m^−2^, preoperative hemoglobin ≤ 10 g·dL^−1^, creatinine ≥ 1.1 mg·dL^−1^, general anesthesia without intraoperative epidural analgesia, and C-reactive protein ≥ 75 mcg·L^−1^. In the multivariate analysis performed after this categorization, the non-use of intraoperative epidural analgesia emerged as the sole variable significantly related to the overall mortality (Table 6).

## 4. Discussion

CS + HIPEC is a long-debated treatment in oncologic surgery. In our sample, the incidence of complications was high, and the postoperative recovery was slow. However, these factors did not contribute to an increased one-year or overall mortality rate. This analysis of the postoperative evolution of all the patients who underwent CS + HIPEC for tumors of different origins over more than 7 years at a tertiary university hospital showed that postoperative complications occurred in 62.3% of cases, with a 12-month mortality rate of 33.8% and a median survival of 63.11 months.

The published data on morbidity following CS + HIPEC vary, ranging from less than 20% to more than 50% [14,15,16,17,18,19,20]. This variability may be because the occurrence of complications depends on the primary tumor causing the peritoneal metastases [2], as well as on the timing of the analysis. Most studies focus on CS + HIPEC procedures for a single tumor origin. In our study, we included interventions for peritoneal metastases from various primary tumors. Furthermore, our analysis accounted for any type of postoperative complication, regardless of its severity, which may have contributed to the high postoperative morbidity rate observed.

The most frequently reported postoperative complications after this surgery are anastomotic dehiscence, intra-abdominal abscess, ileus, and nausea/vomiting [15,17]. Moreover, the combination of hyperthermia and chemotherapy used during HIPEC may disrupt the healing process, increasing the incidence of anastomotic leaks, bleeding, and postoperative small-bowel fistulas [21]. The risk of these complications should be minimized by a thorough lysis of adhesions and the careful performance of anastomosis. Adynamic ileus is the most common morbidity observed postoperatively, although probably one of the least serious complications reported [22]. The main complications detected in our sample were infectious. Septic shock is the main cause of readmission to the ICU in these patients, with sepsis of abdominal origin the most common cause [23]. HIPEC associated with hyperthermia induces apoptosis, inhibits angiogenesis, and promotes protein denaturation [24]. These effects are beneficial in preventing tumor progression but can be harmful from an immunological point of view. Other frequently described complications include kidney failure [25,26], liver dysfunction, myelosuppression [26], bleeding [27], or venous thromboembolisms [28]. Although our analysis did not differentiate the severity of the complications experienced, patients who had at least one complication had a significantly longer hospital stay, which may have affected the overall treatment outcome [6]. Other studies have shown that, although more than 50% of patients experienced severe complications (Clavien–Dindo > 2), the mortality rate was 0% [17].

Patients undergoing CS + HIPEC often suffer from multiple comorbidities, with a worse ASA physical status [2]. These conditions may make them more susceptible to postoperative complications. We did not detect that the ASA physical status or the presence of comorbidities influenced the occurrence of postoperative complications. We also did not detect a relationship between the preoperative hemoglobin and creatinine values and the manifestation of the different complications. However, age [29], male gender [30], preoperative frailty [8], preoperative heart diseases [31], smoking [30], preoperative glycemia [32], intraoperative vasoactive support [23], and intraoperative infusion greater than 6 L of crystalloids [33] have been associated with postoperative complications. Although inflammatory markers are normally elevated intraoperatively and decrease during the first postoperative day [23], C-reactive protein values in the immediate postoperative period were significantly higher in patients who suffered respiratory failure. Postoperative creatinine in our analysis was only detected as a marker of respiratory failure, despite the fact that it is usually a marker of postoperative renal failure. On the other hand, the postoperative need for non-invasive mechanical ventilation and vasoactive support was significantly higher in patients with renal failure and infectious complications, while the need for renal replacement therapies was significantly higher in patients with renal or respiratory failure and infectious complications.

The duration of surgery was significantly longer in patients who subsequently suffered renal failure, adynamic ileus, and any infectious complications in our sample. These data are consistent with those found in previous studies [30,33]. However, the length of surgery did not seem to affect the occurrence of respiratory complications, despite being a risk factor for postoperative pulmonary complications in predictive scores widely used in routine clinical practice [34].

The reported mortality after CS + HIPEC ranges from 0% [17] to 1.6% [35] and 2.1% [19,20]. The same occurs for postoperative mortality and for complications: the mortality rate depends on the time at which the postoperative analysis is performed. Therefore, the variability between the different studies found is notable: it may be around 2% at one month [15,20] and 13.9% at 12 months [35]. A meta-analysis carried out on 16 studies detected survival rates of 41.2–100% at one year, 5.9–87.9% at three years, and around 27–62% at five years postoperatively [26]. The mortality detected in our sample was 10.4% at 6 months and 15.6% at 12 months, and the overall mortality (during the 7 years of this study) was 33.8%. A meta-analysis carried out in epithelial ovarian cancer shows an overall survival after surgery of around 26.6–71.3 months [36], while in colorectal cancer the median overall survival was 41 months, in gastric cancer 14 months, and in mesothelioma 66 months [19]. The median survival in our sample was 63.11 months, despite the inclusion of different primary tumors.

The occurrence of complications was not associated with an increase in postoperative mortality at 1 year. Complications did not correlate with postoperative mortality, according to the findings in reference [17]. Factors contributing to 1-year mortality included lower body mass index (BMI), reduced preoperative hemoglobin and creatinine levels, absence of intraoperative epidural analgesia, and elevated postoperative hemoglobin and C-reactive protein levels. Preoperative optimization of the patient both nutritionally and analytically may offer improvements in postoperative outcomes [10].

Surgical damage associated with chemotherapy treatment can lead to central and peripheral inflammation [37]. The associated postoperative pain is moderate–severe [37], and the use of thoracic epidural analgesia is recommended to reduce respiratory complications and adynamic ileus [23]. Pain assessment with the numerical rating scale in this population yielded values similar to those found in other postsurgical patients [38]. Despite the alterations in coagulation or the appearance of postoperative thrombocytopenia, an increase in the presence of postoperative epidural hematomas has not been detected in these patients [37,39]. The percentage of patients who received intraoperative epidural analgesia was significantly higher among patients who survived. However, no differences were detected in postoperative analgesic management according to the analgesic protocol administered, whether epidural or morphine, and no differences were detected in analgesic management [40] and in the appearance of respiratory complications or adynamic ileus.

### Limitations

Despite the exhaustive collection of data performed in this retrospective analysis, we acknowledge some limitations. The main limitation is the potential loss of patients or data. However, the patients were collected from an electronic database, and the data were recorded exhaustively therein during those seven years. In addition, all the patients operated on during this period were included, which ensures good internal validity. Although multiple types of cancers during a significant time interval were included, it may not be possible to extrapolate the results to other types of centers or to patients with other characteristics. Another limitation was the possibility that there may have been data that were not considered, and that could have influenced the results. Some of these variables not included in the analysis may be related to the nonuse of intraoperative epidural analgesia. We believe that the data analyzed can be easily collected and generally obtained in most centers in which this surgery occurs, making the analysis relevant. In addition, the analysis was carried out according to the data collected from the electronic medical record. Thus, we are unaware of aspects that may be key to understanding, such as the patient’s state of frailty or quality of life. Finally, most of the studies we found were retrospective analyses or meta-analyses, and there was little published information on perioperative risk factors that influence postoperative mortality. Therefore, we were not able to compare our findings with those published in other centers.

## 5. Conclusions

The aggressiveness and duration of CS + HIPEC leads to a high incidence of postoperative complications. However, the appearance of postoperative complications does not seem to affect postoperative survival. There are many perioperative factors that can favor the appearance of postoperative complications and lengthen the hospital stay. The optimization of these parameters, such as preoperative hemoglobin and creatinine, or the careful management of intraoperative analgesia can prevent postoperative complications and speed up patient recovery. In addition, close postoperative monitoring of patients’ evolution and analytical markers can lead to an active approach to prevent the progression of complications.

## Figures and Tables

**Figure 1 healthcare-13-00808-f001:**
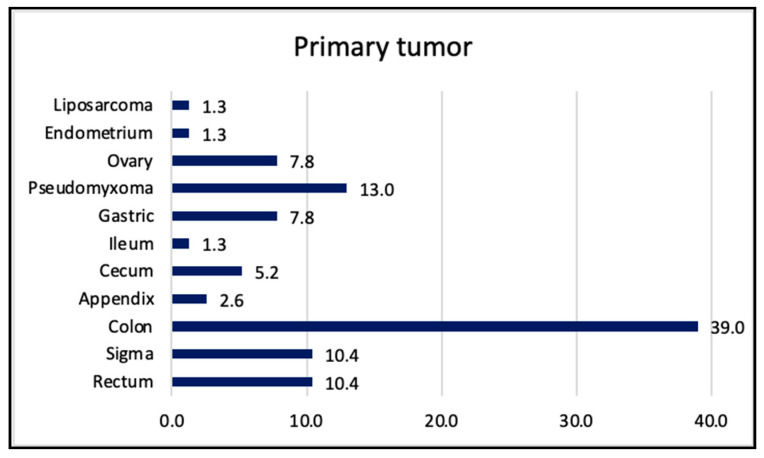
Primary tumor. Data are expressed in relative frequencies.

**Figure 2 healthcare-13-00808-f002:**
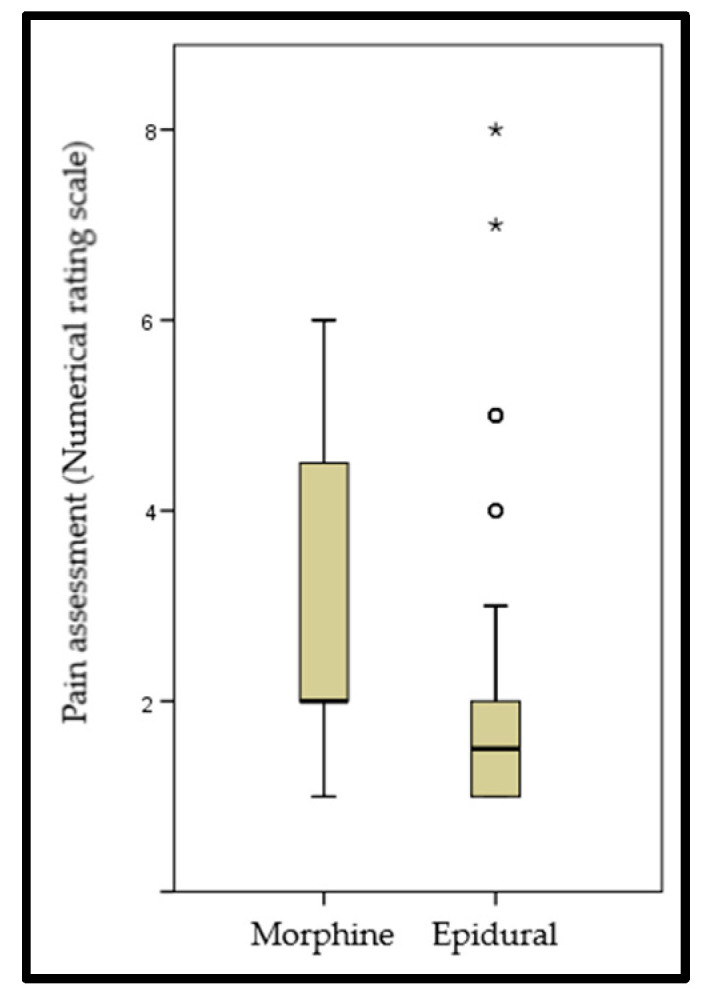
Postoperative pain assessment according to analgesic protocol. Data are expressed as median (IQR). ^O^ and *: outliers.

**Figure 3 healthcare-13-00808-f003:**
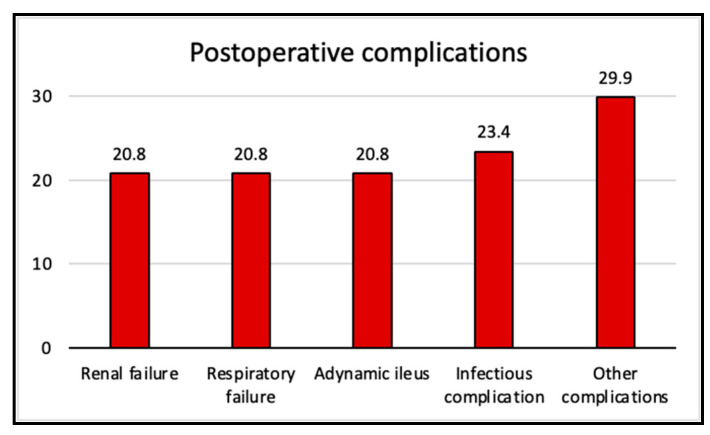
Postoperative complications. Data are expressed as relative frequencies.

**Figure 4 healthcare-13-00808-f004:**
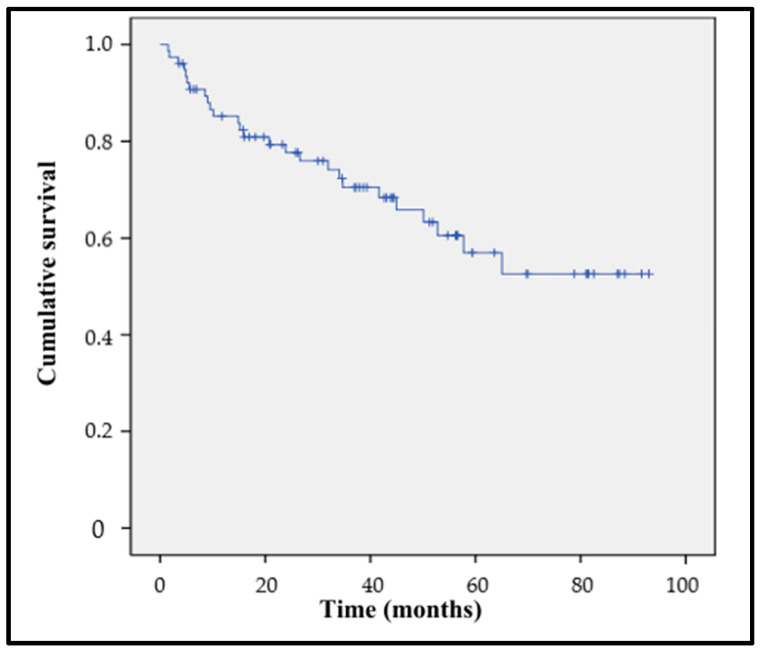
Cumulative postoperative survival.

**Table 1 healthcare-13-00808-t001:** Patient and intraoperative characteristics.

		n = 77
Female gender, n (%)		46 (59.7)
Age, years		61.6 ± 8.7
Height, m		1.66 ± 0.09
Weight, kg		73.2 ± 14.3
Body mass index, kg·m^−2^		26.37 ± 4.25
ASA	I, n (%)	1 (1.3)
	II, n (%)	10 (12.9)
	III, n (%)	38 (49.4)
	IV, n (%)	28 (36.4)
Comorbidities	Arterial hypertension, n (%)	31 (40.3)
	Diabetes mellitus, n (%)	14 (18.2)
	Lung disease, n (%)	7 (9.1)
	Heart diseases, n (%)	6 (7.8)
	Chronic kidney disease, n (%)	2 (2.6)
Preoperative laboratory tests	Hemoglobin, g·dL^−1^	12.5 ± 1.6
	Creatinine, mg·dL^−1^	0.79 ± 0.19
Intraoperative management	Epidural analgesia, n (%)	58 (75.3)
	Transfusions, n (%)	13 (16.9)
	Vasoactive support, n (%)	33 (42.9)
Duration of surgery, min		477 ± 177
Immediate postoperative extubation in the operating room, n (%)		68 (88.3)

Data are expressed as mean ± SD or absolute and relative frequencies. ASA: American Society of Anesthesiologists physical status.

**Table 2 healthcare-13-00808-t002:** Postoperative evolution.

		n = 77
Postoperative therapies	Vasoactive support, n (%)	22 (28.6)
	Non-invasive mechanical ventilation, n (%)	12 (15.6)
	Renal replacement therapies, n (%)	4 (5.2)
Immediate postoperative laboratory tests	Hemoglobin, g·dL^−1^	10.9 ± 1.9
	Creatinine, mg·dL^−1^	0.72 ± 0.31
	Lactate, mmol·L^−1^	1.99 ± 1.69
	C-reactive protein, mcg·L^−1^	81.61 ± 60.69
Pain management	Epidural analgesia, n (%)	62 (80.5)
	Morphine, n (%)	15 (19.5)
Pain subjective assessment	Excellent, n (%)	59 (76.6)
	Fair, n (%)	16 (20.8)
	Bad, n (%)	2 (2.6)
Numerical rating scale, score		2.32 ± 1.68
Complications related to pain management	Poor pain control, n (%)	7 (9.1)
	Paresthesia, n (%)	5 (6.5)
	Arterial hypotension, n (%)	5 (6.5)
	Pruritus, n (%)	1 (1.3)

Data are expressed as mean ± SD or absolute and relative frequencies.

**Table 3 healthcare-13-00808-t003:** Relationship between pre- and intraoperative variables and main postoperative complications.

	Renal Failure			Respiratory Failure			Adynamic Ileus			Infectious Complication		
	No (n = 61)	Yes (n = 16)	*p*	No (n = 61)	Yes (n = 16)	*p*	No (n = 61)	Yes (n = 16)	*p*	No (n = 59)	Yes (n = 18)	*p*
Female gender, n (%)	34 (55.7)	12 (75)	0.162	39 (63.9)	7 (43.7)	0.143	34 (55.7)	12 (75)	0.162	37 (62.7)	9 (50)	0.336
Age, years	61.3 ± 8.0	62.8 ± 11.3	0.549	62.3 ± 7.6	59.1 ± 12.2	0.198	61.2 ± 8.7	63.4 ± 8.8	0.357	61.3 ± 9.0	62.8 ± 7.8	0.510
Body mass index, kg·m^−2^	26.29 ± 4.19	26.68 ± 4.57	0.743	26.09 ± 4.12	27.45 ± 4.67	0.256	26.56 ± 4.16	25.67 ± 4.61	0.462	26.37 ± 4.25	26.37 ± 4.35	0.997
ASA = 4, n (%)	23 (37.7)	5 (31.2)	0.633	23 (37.7)	5 (31.2)	0.633	23 (37.7)	5 (31.2)	0.633	22 (37.3)	6 (33.3)	0.760
Arterial hypertension, n (%)	27 (44.3)	4 (25)	0.162	24 (39.3)	7 (43.7)	0.749	27 (44.3)	4 (25)	0.162	24 (40.7)	7 (38.9)	0.892
Diabetes mellitus, n (%)	13 (21.3)	1 (6.2)	0.164	10 (16.4)	4 (25.0)	0.427	12 (19.7)	2 (12.5)	0.508	12 (20.3)	2 (11.1)	0.374
Lung disease, n (%)	4 (6.6)	3 (18.7)	0.131	5	2	0.594	3 (4.9)	4 (25)	0.013	6 (10.2)	1 (5.5)	0.551
Heart diseases, n (%)	5 (8.2)	1 (6.2)	0.796	3 (4.9)	3 (18.7)	0.066	5 (8.2)	1 (6.2)	0.796	4 (6.8)	2 (11.1)	0.548
Chronic kidney disease, n (%)	2 (3.3)	0 (0)	0.463	2 (3.3)	0 (0)	0.463	2 (3.3)	0 (0)	0.463	2 (3.4)	0 (0)	0.429
Hemoglobin, g·dL^−1^	12.4 ± 1.7	12.9 ± 1.8	0.340	12.5 ± 1.7	12.6 ± 1.9	0.737	12.5 ± 1.8	12.5 ± 1.7	0.934	12.6 ± 1.8	12.3 ± 1.7	0.584
Creatinine, mg·dL^−1^	0.81 ± 0.21	0.73 ± 0.121	0.197	0.79 ± 0.21	0.80 ± 0.15	0.813	0.81 ± 0.21	0.72 ± 0.15	0.128	0.79 ± 0.21	0.81 ± 0.13	0.674
Epidural analgesia, n (%)	44 (72.1)	14 (87.5)	0.204	45 (73.8)	13 (81.2)	0.537	44 (72.1)	14 (87.5)	0.204	45 (76.3)	13 (72.2)	0.727
Transfusions, n (%)	8 (13.1)	5 (31.2)	0.085	10 (16.4)	3 (18.7)	0.823	10 (16.4)	3 (18.7)	0.823	8 (13.5)	5 (27.8)	0.159
Vasoactive support, n (%)	24 (39.3)	9 (56.2)	0.224	26 (42.6)	7 (43.7)	0.935	26 (42.6)	7 (43.7)	0.935	25 (42.4)	8 (44.4)	0.876
Duration of surgery, min	455 ± 183	566 ± 117	0.029	459 ± 169	544 ± 197	0.089	443 ± 166	604 ± 163	0.001	457 ± 189	541 ± 110	0.024
Extubation in the OR, n (%)	56 (91.8)	12 (75)	0.063	55 (90.2)	13 (81.2)	0.323	55 (90.2)	13 (81.2)	0.323	53 (89.8)	15 (83.3)	0.453

Data are expressed as mean ± SD or absolute and relative frequencies.

**Table 4 healthcare-13-00808-t004:** Relationship between postoperative variables and main postoperative complications.

	Renal Failure			Respiratory Failure			Adynamic Ileus			Infectious Complication		
	No (n = 61)	Yes (n = 16)	*p*	No (n = 61)	Yes (n = 16)	*p*	No (n = 61)	Yes (n = 16)	*p*	No (n = 59)	Yes (n = 18)	*p*
Post-vasoactive support, n (%)	14 (22.9)	8 (50)	0.033	16 (26.2)	6 (37.5)	0.374	15 (24.6)	7 (43.7)	0.131	14 (23.7)	8 (44.4)	0.089
Post-NIMV, n (%)	6 (9.8)	6 (37.5)	0.007	7 (11.5)	5 (31.2)	0.052	8 (13.1)	4 (25)	0.243	6 (10.2)	6 (33.3)	0.018
Post-RRTs, n (%)	1 (1.6)	3 (18.7)	0.006	1 (1.6)	3 (18.7)	0.006	2 (3.3)	2 (12.5)	0.139	1 (1.7)	3 (16.7)	0.012
Post-hemoglobin, g·dL^−1^	11.1 ± 1.8	10.4 ± 1.8	0.161	10.8 ± 1.8	11.6 ± 1.9	0.140	11.16 ± 1.83	10.26 ± 1.92	0.088	11.17 ± 1.94	10.33 ± 1.47	0.098
Post-creatinine, mg·dL^−1^	0.71 ± 0.24	0.78 ± 0.48	0.375	0.67 ± 0.21	0.93 ± 0.48	0.001	0.73 ± 0.24	0.71 ± 0.49	0.826	0.69 ± 0.26	0.81 ± 0.43	0.164
Lactate, mmol·L^−1^	1.86 ± 1.66	2.48 ± 1.79	0.196	1.88 ± 1.71	2.39 ± 1.63	0.288	2.03 ± 1.74	1.81 ± 1.55	0.643	1.96 ± 1.74	2.06 ± 1.60	0.831
C-reactive protein, mcg·L^−1^	84.1 ± 66.3	71.6 ± 27.7	0.481	74.6 ± 49.4	108.0 ± 88.8	0.049	80.7 ± 66.9	84.9 ± 27.9	0.807	79.7 ± 62.2	87.9 ± 56.7	0.619
Analgesic protocol: Morphine, n (%)	11 (18)	4 (25)	0.531	14 (22.9)	1 (6.2)	0.133	10 (16.4)	5 (31.2)	0.182	11 (18.6)	4 (22.2)	0.737
NRS, score	2.3 ± 1.7	2.4 ± 1.6	0.894	2.5 ± 1.8	1.7 ± 1.2	0.060	2.2 ± 1.6	2.9 ± 1.9	0.142	2.4 ± 1.7	2.2 ± 1.5	0.652

Data are expressed as mean ± SD or absolute and relative frequencies. NIMV: non-invasive mechanical ventilation; RRTs: renal replacement therapies; NRS: numerical rating scale.

**Table 5 healthcare-13-00808-t005:** Relationship between pre-, intra-, and postoperative variables and mortality.

	Mortality 12 Months				Overall Mortality			
	No (n = 65)	Yes (n = 12)	OR (95% CI) or md (95% CI)	*p*	No (n = 51)	Yes (n = 26)	OR (95% CI) or md (95% CI)	*p*
Female gender, n (%)	24 (36.9)	7 (58.3)	2.4 (0.7–8.4)	0.165	33 (64.7)	13 (50)	1.8 (0.7–4.8)	0.213
Age, years	61.5 ± 9.0	62.6 ± 7.2	1.1 (−4.4–6.6)	0.686	61.0 ± 9.8	63 ± 6	1.9 (−2.3–6.1)	0.374
Body mass index, kg·m^−2^	26.84 ± 4.41	23.85 ± 1.84	−2.9 (−5.6–−0.4)	<0.001	27.01 ± 4.64	25.12 ± 3.03	−1.9 (−3.6–−0.1)	0.035
ASA = 4, n (%)	26 (40)	2 (16.7)	0.3 (0.1–1.5)	0.123	20 (39.2)	8 (30.8)	0.7 (0.2–1.9)	0.466
Arterial hypertension, n (%)	28 (43.1)	3 (25)	0.4 (0.1–1.8)	0.241	23 (45.1)	8 (30.8)	0.5 (0.2–1.5)	0.225
Diabetes mellitus, n (%)	13 (20)	1 (8.3)	0.4 (0.0–3.1)	0.336	9 (17.6)	5 (19.2)	1.1 (0.3–3.7)	0.865
Preop hemoglobin, g·dL^−1^	12.79 ± 1.59	11.06 ± 1.96	−1.7 (−2.8–−0.7)	0.001	12.88 ± 1.69	11.79 ± 1.68	−1.1 (−1.9–−0.3)	0.010
Preop creatinine, mg·dL^−1^	0.81 ± 0.2	0.69 ± 0.12	−0.1 (−0.2–−0.0)	0.007	0.83 ± 0.2	0.72 ± 0.17	−0.1 (−0.2–−0.0)	0.024
Epidural analgesia, n (%)	52 (80)	6 (50)	0.2 (0.1–0.9)	0.027	42 (82.3)	16 (61.5)	0.3 (0.1–0.9)	0.045
Transfusions, n (%)	11 (16.9)	2 (16.7)	0.9 (0.2–5.1)	0.983	8 (15.7)	5 (19.2)	1.3 (0.4–4.4)	0.695
Vasoactive support, n (%)	29 (44.6)	4 (33.3)	0.6 (0.2–2.3)	0.468	22 (43.1)	11 (42.3)	0.9 (0.4–2.5)	0.945
Extubation in the OR, n (%)	57 (87.7)	11 (91.7)	1.5 (0.2–13.6)	0.694	45 (88.2)	23 (88.5)	1.0 (0.2–4.6)	0.977
Post-vasoactive support, n (%)	21 (32.3)	1 (8.3)	0.2 (0.0–1.6)	0.091	15 (29.4)	7 (26.9)	0.9 (0.3–2.5)	0.819
Post-NIMV, n (%)	9 (13.8)	3 (25)	2.1 (0.5–9.1)	0.328	8 (15.7)	4 (15.4)	0.9 (0.3–3.6)	0.972
Post-RRTs, n (%)	4 (6.1)	0 (0)	-	0.377	3 (5.9)	1 (3.8)	0.6 (0.1–6.5)	0.703
Hemoglobin, g·dL^−1^	10.79 ± 1.57	11.94 ± 2.92	1.1 (0.0–2.3)	0.049	10.81 ± 1.48	11.29 ± 2.46	0.5 (−0.4–1.4)	0.284
Creatinine, mg·dL^−1^	0.71 ± 0.91	0.77 ± 0.31	0.1 (−0.1–0.2)	0.558	0.83 ± 0.2	0.72 ± 0.28	−0.0 (−0.1–0.1)	0.987
Lactate, mmol·L^−1^	1.87 ± 1.55	2.58 ± 2.31	0.7 (−0.3–1.8)	0.184	1.87 ± 1.59	2.21 ± 1.89	0.3 (−0.5–1.1)	0.413
C-reactive protein, mcg·L^−1^	72.9 ± 49.4	127.6 ± 91.5	54.6 (18.5–90.8)	0.004	69.0 ± 40.6	107.2 ± 84.0	38.2 (9.8–66.6)	0.040
Analgesic protocol: Morphine, n (%)	13 (20)	2 (16.7)	1.2 (0.2–6.4)	0.789	10 (19.6)	5 (19.2)	1.0 (0.3–3.4)	0.968
NRS, score	2.4 ± 1.7	1.8 ± 1.2	−0.5 (−1.6–0.5)	0.273	2.4 ± 1.8	2.1 ± 1.4	−0.3 (−1.1–0.5)	0.528

Data are expressed as mean ± SD or absolute and relative frequencies. ASA: American Society of Anesthesiologists physical status; OR: operating room; NIMV: non-invasive mechanical ventilation; RRTs: renal replacement therapies; NRS: numerical rating scale.

**Table 6 healthcare-13-00808-t006:** Multivariate analysis of overall mortality prediction.

		*p*	Exp (B)	95% CI of Exp (B)
1st Step	BMI > 30 kg·m^−2^	0.304	0.464	0.108–2.003
	Preoperative hemoglobin < 10 g·dL^−1^	0.945	0.936	0.142–6.181
	Preoperative creatinine > 1.1 mg·dL^−1^	0.402	0.373	0.037–3.747
	Non-intraoperative epidural analgesia	0.049	3.146	1.004–9.861
	C-reactive protein > 75 mcg·L^−1^	0.097	2.445	0.850–7.026
	Constant	0.005	0.281	
2nd Step	BMI > 30 kg·m^−2^	0.304	0.467	0.110–1.991
	Preoperative creatinine > 1.1 mg·dL^−1^	0.404	0.376	0.038–3.738
	Non-intraoperative epidural analgesia	0.047	3.126	1.013–9.647
	C-reactive protein > 75 mcg·L^−1^	0.091	2.424	0.867–6.780
	Constant	0.005	0.281	
3rd Step	BMI > 30 kg·m^−2^	0.350	0.503	0.119–2.124
	Non-intraoperative epidural analgesia	0.053	3.002	0.988–9.123
	C-reactive protein > 75 mcg·L^−1^	0.081	2.488	0.895–6.918
	Constant	0.003	0.260	
4th Step	Non-intraoperative epidural analgesia	0.040	3.193	1.054–9.676
	C-reactive protein > 75 mcg·L^−1^	0.066	2.592	0.940–7.149
	Constant	0.000	0.223	
5th Step	Non-intraoperative epidural analgesia	0.039	3.111	1.060–9.128
	Constant	0.001	0.357	

## Data Availability

The data presented in this study are available on request from the corresponding author.

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
