# Peer review of "Cytoreductive Surgery (CS) with Hyperthermic Intraperitoneal Chemotherapy (HIPEC): Postoperative Evolution, Adverse Outcomes and Perioperative Risk Factors"

_healthcare, 2025, doi:10.3390/healthcare13070808_

Round 1
Reviewer 1 Report
Comments and Suggestions for Authors
Relevant manuscript, but I think it should be restructured.
The abstract mentions differences, but it does not really say whether they increase or decrease or the values that can guide a result.
I think that the phrase “the most frequent complication” should be used only once. “The most frequent respiratory complication was pleural effusion in 4 patients (5.2%). The most frequent complication was infection, diag-134 nosed in 23.4% of cases (18 patients)”
The manuscript is interesting, with very useful data for clinicians and a basis for future studies. However, the order and analysis as it is presented does not facilitate its understanding. Although all the data are presented, it is complex to read. I think that the analysis and results should be more organized and structured, divided by sections. The interesting thing would be the associations with mortality, or associations with complications. It is suggested that quantitative variables also be analyzed as associations, dichotomizing them through relevant clinical cut-off points, or through cut-off points obtained through ROC curves. With this, the summary will be easier to elaborate, through association results. With a new way of analyzing the results, perhaps the focus of the article can be centered on complications and their associated factors, although that is a point that must be determined. With the current manuscript, a detailed description is observed, but it does not leave a clear message about what they want to describe or the main findings. I consider that the manuscript is relevant, but it should be restructured.
Author Response
Q1) Relevant manuscript, but I think it should be restructured. The abstract mentions differences, but it does not really say whether they increase or decrease or the values that can guide a result.
R1.- We truly appreciate this comment. We have added the mean differences and the Odds ratios to the abstract to make the associations clearer:
Renal failure was related to length of surgery [mean difference (md) 111 min, CI 95% 11 – 210, p = 0.029], postoperative vasoactive support [Odds Ratio (OR) 3.4, CI 95% 1.1 – 10.6, p = 0.033], and non-invasive mechanical ventilation (OR 5.5, CI 95% 1.5 – 20.5, p = 0.007). Respiratory failure was associated to renal replacement therapies (OR 13.8, CI 95% 1.3 – 143.9, p = 0.006), postoperative creatinine (md 0.27 mg·dl-1, CI 95% 0.1 – 0.4, p = 0.001) and C-reactive protein (md 33.5 mcg·l-1, CI 95% 0.1 – 66.8, p = 0.049). Infectious complications were related to length of surgery (md 84 min, CI 95% 12 – 156, p = 0.024), non-invasive mechanical ventilation (OR 4.4, CI 95% 1.2 – 16.1, p = 0.018) and renal replacement therapies (OR 11.6, CI 95% 1.1 – 119.6, p = 0.012). Hospital stay was longer in patients with complications (md 14.8 + 5.5 days, CI 95% 3.8 – 25.8, p = 0.009). Mortality rate at 12-months was 15.6%. Mortality risk factors were preoperative hemoglobin (md -1.7 g·dl-1, CI 95% -2.8 – -0.7, p = 0.001), creatinine (md -0.12 mg·dl-1, CI 95% -0.21 – -0.04, p = 0.007), postoperative hemoglobin (md -1.15 g·dl-1, CI 95% 0.01 – 2.30, p = 0.049) and C-reactive protein (md 54.6 mcg·l-1, CI 95% 18.5 – 90.8, p = 0.004). Intraoperative epidural analgesia was found to be a protective factor for 12-month mortality (OR 0.25, CI 95% 0.07 – 0.90 p = 0.027).
Q2) I think that the phrase “the most frequent complication” should be used only once. “The most frequent respiratory complication was pleural effusion in 4 patients (5.2%). The most frequent complication was infection, diag-134 nosed in 23.4% of cases (18 patients)”
R2.- We fully agree with this comment. We have made the requested changes, leaving only one “most frequent complication”.
Q3) The manuscript is interesting, with very useful data for clinicians and a basis for future studies. However, the order and analysis as it is presented does not facilitate its understanding. Although all the data are presented, it is complex to read. I think that the analysis and results should be more organized and structured, divided by sections. The interesting thing would be the associations with mortality, or associations with complications.
R3.- We deeply appreciate the Reviewer’s positive comments on the manuscript. We have restructured the Results section to make it easily readable. We have also added the following paragraph to clarify quantitative results:
Patients who suffered from renal failure showed longer duration of surgery (md 111 min, CI 95% 11 – 210, p = 0.029), and more use of postoperative vasoactive support (OR 3.4, CI 95% 1.1 – 10.6, p = 0.033) and noninvasive mechanical ventilation (OR 5.5, CI 95% 1.5 – 20.5, p = 0.007). Respiratory failure was associated with postoperative renal replacement therapies (OR 13.8, 95% CI 1.3 – 143.9, p = 0.006) and with higher postoperative values of creatinine (md 0.27 mg·dl-1, 95% CI 0.1 – 0.4, p = 0.001) and C-reactive protein (md 33.5 mcg·l-1, 95% CI 0.1 – 66.8, p = 0.049). Patients who suffered infectious complications also showed longer duration of surgery (md 84 min, 95% CI 12 – 156, p = 0.024) and higher postoperative use of non-invasive mechanical ventilation (OR 4.4, 95% CI 1.2 – 16.1, p = 0.018) and renal replacement therapies (OR 11.6, 95% CI 1.1 – 119.6, p = 0.012).
Q4) It is suggested that quantitative variables also be analyzed as associations, dichotomizing them through relevant clinical cut-off points, or through cut-off points obtained through ROC curves. With this, the summary will be easier to elaborate, through association results. With a new way of analyzing the results, perhaps the focus of the article can be centered on complications and their associated factors, although that is a point that must be determined. With the current manuscript, a detailed description is observed, but it does not leave a clear message about what they want to describe or the main findings. I consider that the manuscript is relevant, but it should be restructured.
R4.- We thank the Reviewer for this suggestion. We have structured the Results section to make it easier to read. We have performed the recommended dichotomization of quantitative variables related to overall mortality and the corresponding multivariate analysis, obtaining the following:
Quantitative variables significantly related to overall mortality were dichotomized through relevant clinical cut-off points as follows: BMI > 30 kg·m-1, preoperative hemoglobin < 10 g·dl-1, creatinine > 1.1 mg·dl-1, general anesthesia without intraoperative epidural analgesia, and C-reactive protein > 75 mcg·l-1. In the multivariate analysis performed after this categorization, the non-use of intraoperative epidural analgesia emerged as the sole significant variable (p = 0.039) related to overall mortality (Table 6).
Reviewer 2 Report
Comments and Suggestions for Authors
The authors present a study on the perioperative factors and their impact on postoperative mortality in patients undergoing CS and HIPEC. The study comprises multiple types of cancers with peritoneal carcinomatosis, treated in a tertiary level hospital, across a significant time interval. The data focuses on correlations observed between multiple pre- and intraoperative elements in relation to postoperative complocations and mortality. The text is well-written, with detailed information on the data collected. The results are presented in with multiple graphs and figures in an effort to more clearly provide the reader with the observed data. However, the vast amount of data presented with some of the tables may be confusing and hard to follow. Although, the authors provide explanations for several limitations of this study, in my opinion one important aspect that is missing, namely a multivariate analysis of the data that were observed as significant in the univariate analysis. Given that the data is already available, this can be further added to the study, refining the results and reducing the effect of confounding factors. Should the authors add this part to the results section, a further revision of the discussions section would be warranted.
Author Response
Q1) The authors present a study on the perioperative factors and their impact on postoperative mortality in patients undergoing CS and HIPEC. The study comprises multiple types of cancers with peritoneal carcinomatosis, treated in a tertiary level hospital, across a significant time interval. The data focuses on correlations observed between multiple pre- and intraoperative elements in relation to postoperative complications and mortality. The text is well-written, with detailed information on the data collected. The results are presented in with multiple graphs and figures in an effort to more clearly provide the reader with the observed data.
R1.- We really appreciate the Reviewer's positive comments on the manuscript.
Q2) However, the vast amount of data presented with some of the tables may be confusing and hard to follow. Although, the authors provide explanations for several limitations of this study, in my opinion one important aspect that is missing, namely a multivariate analysis of the data that were observed as significant in the univariate analysis. Given that the data is already available, this can be further added to the study, refining the results and reducing the effect of confounding factors. Should the authors add this part to the results section, a further revision of the discussions section would be warranted.
R3.- We thank the Reviewer for the recommendations. As requested, we have restructured the Results section and added information to clarify quantitative results.
Patients who suffered from renal failure showed longer duration of surgery (md 111 min, CI 95% 11 – 210, p = 0.029), and greater use of postoperative vasoactive support (OR 3.4, CI 95% 1.1 – 10.6, p = 0.033) and noninvasive mechanical ventilation (OR 5.5, CI 95% 1.5 – 20.5, p = 0.007). Respiratory failure was associated with postoperative renal replacement therapies (OR 13.8, 95% CI 1.3 – 143.9, p = 0.006) and with higher postoperative values of creatinine (md 0.27 mg·dl-1, 95% CI 0.1 – 0.4, p = 0.001) and C-reactive protein (md 33.5 mcg·l-1, 95% CI 0.1 – 66.8, p = 0.049). Patients who suffered infectious complications also showed longer duration of surgery (md 84 min, 95% CI 12 – 156, p = 0.024) and higher postoperative use of non-invasive mechanical ventilation (OR 4.4, 95% CI 1.2 – 16.1, p = 0.018) and renal replacement therapies (OR 11.6, 95% CI 1.1 – 119.6, p = 0.012).
We have also performed a multivariate analysis to predict the overall mortality:
Quantitative variables significantly related to overall mortality were dichotomized through relevant clinical cut-off points as follows: BMI > 30 kg·m-1, preoperative hemoglobin < 10 g·dl-1, creatinine > 1.1 mg·dl-1, general anesthesia without intraoperative epidural analgesia, and C-reactive protein > 75 mcg·l-1. In the multivariate analysis performed after this categorization, the non-use of intraoperative epidural analgesia emerged as the sole significant variable (p = 0.039) related to overall mortality (Table 6).
Consequently, the Discussion section has been modified.
Reviewer 3 Report
Comments and Suggestions for Authors
The present study focuses on the postoperative outcomes following cytoreductive surgery (CS) and HIPEC. This is a therapeutical option that has long been debated in the field of surgical oncology, and never truely agreed upon, therefore research undertaken to demonstrate its efficiency or not, is important. The authors performed a retrospective analysis on 77 patients with peritoneal carcinomatosis of different origins.The data is presented well and the authors clearly state the methods used. However, in the limitations section several aspects are not included. These aspects should be mentioned by the authors or, if possible, added in the analysis of the study. Firstly, the authors performed solely a univariate evaluation of the data, whereas I would have expected a multivariate analysis, this would validate or invalidate data found to be statically significant in the univariate analysis. Second, the authors mention that infection was the most often encountered complication in their study sample. In my opinion the authors should elaborate on this particular aspect, with details on type of infection, number of postoperative days until infection occured/was diagnosed, type of antibiotic prophylaxie used. Furthermore, the authors should make a comment on the novel information their study brings, as it does not show any survival benefits (this is of course also hard to support, given the lack of a control group and the heterogeneity of primary tumors included in the study).
Comments on the Quality of English LanguageSome English grammar errors are present throughout the text. A review of the text would greatly improve flow for the readers.
Author Response
Q1) The present study focuses on the postoperative outcomes following cytoreductive surgery (CS) and HIPEC. This is a therapeutical option that has long been debated in the field of surgical oncology, and never truly agreed upon, therefore research undertaken to demonstrate its efficiency or not, is important. The authors performed a retrospective analysis on 77 patients with peritoneal carcinomatosis of different origins. The data is presented well and the authors clearly state the methods used.
R1.- We deeply appreciate the positive comments on the research and the manuscript. Following this comment, we have added the following limitation:
Although multiple types of cancers during a significant time interval were included, it may not be possible to extrapolate the results to other types of centers or to patients with other characteristics.
Q2) However, in the limitations section several aspects are not included. These aspects should be mentioned by the authors or, if possible, added in the analysis of the study. Firstly, the authors performed solely a univariate evaluation of the data, whereas I would have expected a multivariate analysis, this would validate or invalidate data found to be statically significant in the univariate analysis.
R2.- Thank you very much for this comment. Following this recommendation and that made by other reviewers, we have performed a multivariate analysis of the factors that were statistically related to overall mortality in the univariate analysis.
Q3) Second, the authors mention that infection was the most often encountered complication in their study sample. In my opinion the authors should elaborate on this particular aspect, with details on type of infection, number of postoperative days until infection occurred/was diagnosed, type of antibiotic prophylaxis used.
R3.- Thank you very much for this comment. Following the Reviewer’s suggestion, we have increased details according to infectious complications:
The most frequent complication was infection, diagnosed in 23.4% of cases (18 patients): 7 of these patients (9.1% of the total sample) had catheter-associated infection (mean time to diagnosis: 9 + 4 days), while the rest (14.3% of the total sample) experienced abdominal cavity infections: 6 patients (7.8%) suffered intraabdominal collections (mean time to diagnosis: 9 + 3 days), 2 patients (2.6%) presented small bowel fistulae (mean time to diagnosis: 7 + 1 days), 2 patients (2.6%) showed pancreatic fistula (mean time to diagnosis: 8 + 1 days), and only one (1.3%) had anastomotic dehiscence (time to diagnosis: 10 days). All patients had received perioperatively amoxicillin-clavulanic acid as prophylaxis, and then targeted antibiotic treatment after diagnosis of infection.
Q4) Furthermore, the authors should make a comment on the novel information their study brings, as it does not show any survival benefits (this is of course also hard to support, given the lack of a control group and the heterogeneity of primary tumors included in the study).
R4.- We truly appreciate this comment. We have added the following in the first paragraph of the Discussion section in the revised manuscript:
CS + HIPEC is a long-debated treatment in oncologic surgery. In our sample, the incidence of complications is high and the postoperative course is slow. However, these factors do not contribute to an increased one-year or overall mortality rate.
Round 2
Reviewer 1 Report
Comments and Suggestions for Authors
The authors made the suggested changes, and the manuscript is now clearer and easier to understand. I believe it can be accepted in its current form.
Author Response
Comments 1: The authors made the suggested changes, and the manuscript is now clearer and easier to understand. I believe it can be accepted in its current form.
Response 1: We deeply thank the reviewer for the positive comments of the revised manuscript.